# Plasticity of Coagulase-Negative Staphylococcal Membrane Fatty Acid Composition and Implications for Responses to Antimicrobial Agents

**DOI:** 10.3390/antibiotics9050214

**Published:** 2020-04-28

**Authors:** Kiran B. Tiwari, Craig Gatto, Brian J. Wilkinson

**Affiliations:** 1A-TEK Inc., 1430 Spring Hill Road, McLean, VA 22102, USA; 2School of Biological Sciences, Illinois State University, Normal, IL 61761, USA

**Keywords:** branched-chain fatty acids, straight-chain saturated fatty acids, straight-chain unsaturated fatty acids, coagulase-negative staphylococci, membrane fatty acid composition, membrane fluidity, serum, antimicrobial fatty acids, antimicrobial susceptibility

## Abstract

*Staphylococcus aureus* demonstrates considerable membrane lipid plasticity in response to different growth environments, which is of potential relevance to response and resistance to various antimicrobial agents. This information is not available for various species of coagulase-negative staphylococci, which are common skin inhabitants, can be significant human pathogens, and are resistant to multiple antibiotics. We determined the total fatty acid compositions of *Staphylococcus auricularis*, *Staphylococcus capitis*, *Staphylococcus epidermidis*, *Staphylococcus haemolyticus*, *Staphylococcus hominis*, *Staphylococcus saprophyticus*, and *Staphylococcus aureus* for comparison purposes. Different proportions of branched-chain and straight-chain fatty acids were observed amongst the different species. However, growth in cation-supplemented Mueller–Hinton broth significantly increased the proportion of branched-chain fatty acids, and membrane fluidities as measured by fluorescence anisotropy. Cation-supplemented Mueller–Hinton broth is used for routine determination of antimicrobial susceptibilities. Growth in serum led to significant increases in straight-chain unsaturated fatty acids in the total fatty acid profiles, and decreases in branched-chain fatty acids. This indicates preformed fatty acids can replace biosynthesized fatty acids in the glycerolipids of coagulase-negative staphylococci, and indicates that bacterial fatty acid biosynthesis system II may not be a good target for antimicrobial agents in these organisms. Even though the different species are expected to be exposed to skin antimicrobial fatty acids, they were susceptible to the major skin antimicrobial fatty acid sapienic acid (C16:1Δ6). Certain species were not susceptible to linoleic acid (C18:2Δ9,12), but no obvious relationship to fatty acid composition could be discerned.

## 1. Introduction

Species of members of the genus *Staphylococcus* are divided into two major groups based upon their ability to coagulate plasma. By far the best-known coagulase-positive species is *Staphylococcus aureus*, which is a virulent pathogen that is often resistant to multiple antibiotics, including β-lactam antibiotics, often through presence of the *mecA* determinant. In contrast the coagulase-negative staphylococci (CONS) are a large heterogeneous group of fifty-three validly described species, as of 2019 [1], from animal host-associated to environmental species varying in pathogenic potential from considerable down to non-pathogenic. Antibiotic resistance rates continue to increase in CONS [1,2].

Novel antimicrobial agents are needed for the therapy of *S. aureus* infections given resistance to multiple anti-staphylococcal agents exhibited by the organism [3]. The *S. aureus* cytoplasmic membrane plays an important role in the susceptibility and resistance to various antimicrobial agents of the bacterium. The cytoplasmic membrane of *S. aureus* is composed of the phospholipids phosphatidyl glycerol, lysyl-phosphatidyl glycerol and cardiolipin, and the glycolipids monoglucosyldiglyceride and diglucosyldiglyceride [4,5]. Decrease in phosphatidyl glycerol, and increases in lysyl-phosphatidyl glycerol and cardiolipin contents have been associated with decreased susceptibility to the membrane active antimicrobial daptomycin [6]. The fatty acids esterified to the glycerol moiety of these glycerolipids are a mixture of branched-chain fatty acids (BCFAs) and straight-chain fatty acids (SCFAs) biosynthesized by the organism when grown in conventional laboratory media [7,8]. There are two types of BCFA in *S. aureus*, namely iso fatty acids where the methyl branch is on the penultimate carbon of the fatty acid chain, and anteiso fatty acids where the methyl branch is on the antepenultimate carbon. Even-numbered iso fatty acids, odd-numbered iso fatty acids, and odd-numbered anteiso fatty acids are biosynthesized from the branched-chain amino acids valine, leucine, and isoleucine, respectively [9]. BCFAs increase membrane fluidity, whereas SCFAs decrease it [10]. Anteiso fatty acids increase membrane fluidity more than iso fatty acids because they pack into the membrane more loosely [10]. The growth medium has significant effects on the balance of BCFAs and SCFAs, with Mueller–Hinton broth (MHB) resulting in a much higher proportion of BCFAs than a medium such as tryptic soy broth (TSB) [7]. Interestingly, MHB is the recommended medium for routine determination of antibiotic susceptibilities in the clinical microbiology laboratory [11]. The other major *S. aureus* membrane lipid component is the golden yellow carotenoid known as staphyloxanthin [12]. Staphyloxanthin is generally thought to decrease membrane fluidity and impacts *S. aureus* susceptibility to various antimicrobial peptides and fatty acids [13,14]. For the most part, this pigment is absent from coagulase-negative species. Staphyloxanthin production varies with different culture conditions and different *S. aureus* strains [15]. In a carotenoid-deficient mutant, growth in MHB resulted in a more fluid membrane [15].

In laboratory media, the *S. aureus* BCFAs and SCFAs are biosynthesized by the fatty acid biosynthesis system II (FASII) [16]. This pathway is unique to bacteria and considerable effort has been devoted to developing novel antimicrobials that target this pathway [16]. However, the typical fatty acid composition of *S. aureus* grown in laboratory media, characterized by a mixture of BCFAs and SCFAs, is unlikely to mimic that of the organism growing in vivo in an infection. *S. aureus* has the ability, like many pathogenic bacteria, to utilize preformed host fatty acids, typically straight-chain unsaturated fatty acids (SCUFAs) and SCFAs, in what is thought to be a means of saving energy and carbon that would otherwise be devoted to fatty acid biosynthesis [17,18]. Thus, when *S. aureus* is grown in the presence of complex host biological materials such as serum [7], or low-density lipoprotein [19], BCFAs decrease and SCFAs and SCUFAs increase. Fatty acids are incorporated into cellular polar lipids through the activities of the fatty acid kinase system, FakAB, and PlsX and PlsY [20]. It has been proposed that there is a requirement for fatty acid anteiso C15:0 on the *sn*-2 portion of *S. aureus* glycerolipids, and hence a requirement for operation of the FASII system when *S. aureus* is growing in the host. However, demonstration of phosphatidyl glycerol species with no biosynthesized fatty acids on either the *sn*-1 or *sn*-2 position of the glycerol moiety seriously undermines the viability of agents targeting the FASII system [21,22].

Although *S. aureus* can incorporate free fatty acids into lipids from media supplemented with them [20,23], the bacterium has a complex relationship with the variety of SCUFAs it might encounter. Very closely related fatty acid structures can either be inhibitory to growth at low concentrations or can have little effect on growth at relatively high concentrations [24,25,26]. For example, sapienic acid (C16:1Δ6) and palmitoleic acid (C16:1Δ9) are highly inhibitory, whereas oleic acid (C18:1Δ9) and vaccenic acid (C18:1Δ11) are not inhibitory and are actually incorporated into the phospholipids by the pathogen [26]. Sapienic acid (C16:1Δ6) is a major antimicrobial fatty acid of human skin, whereas palmitoleic acid (C16:1Δ9) performs this function in other mammalian species [27,28]. Little information is available on how CONS respond to antimicrobial fatty acids to which they are constantly exposed given that they are prominent members of the skin bacterial flora.

The CONS are a heterogeneous group of organisms colonizing the skin and mucus membranes of human and other animal hosts. Based upon sequences of four loci, Lamers et al. [29] placed species into 15 phylogenetic cluster groups [29]. Different species may have characteristic body ecological niches [30]. *S. epidermidis* is prevalent on moist areas and is the most prevalent staphylococcal species on skin. It was cultured from the nose of 97% of people in a recent study [31]. *S. haemolyticus* and *S. hominis* are associated with areas high in apocrine glands. *S. capitis* is isolated from the forehead and scalp. *S. saprophyticus* colonizes the genitourinary tract and rectum, and is a common cause of urinary tract infection in women [32]. *S. auricularis* is uniquely found in the external ear canal [30]. As a group, the CONS tend to cause infections in immunocompromised patients and those with indwelling biomedical devices given the propensity for these bacteria to form biofilms [1,30].

The well documented plasticity of *S. aureus* membrane lipid composition clearly is an important consideration in antibiotic response and resistance, lack of susceptibility to FASII inhibitors, and susceptibility to antimicrobial skin SCUFAs. In contrast, there is little information on the impact of different media on the fatty acid composition and membrane fluidity of CONS. Fatty acids in the CONS are a mixture of BCFAs and SCFAs with differences in the proportions of individual fatty acids such as anteiso C15:0, anteiso C17:0, iso C15:0, C18:0 and C20:0 amongst individual species [33,34]. 

In this study, we investigate the effect of growth medium on the fatty acid composition and membrane fluidity of CONS. We also investigated the response of the different species to host-derived antimicrobial SCUFAs, sapienic acid, palmitoleic acid, and linoleic acid. As in *S. aureus*, growth in MHB increased the proportion of BCFAs and membrane fluidity in CONS. CONS lack staphyloxanthin, which removes a complication from the determination of membrane fluidity. The different species had the ability to incorporate SCFAs and SCUFAs from serum. 

## 2. Results and Discussions

### 2.1. Different Proportions of Branched-Chain Fatty Acids and Straight-Chain Fatty Acids in Different CONS and Increase in Branched-Chain Fatty Acids when Grown in Mueller-Hinton Broth

To investigate the variability of fatty acid compositions in the CONS, they were grown in TSB, MHB and the fatty acid compositions were analyzed by gas-liquid chromatography of fatty acid methyl esters. A strain of *S. aureus* was also used for comparative purposes. The different species were observed to have different proportions of fatty acid compositions in a given culture medium. TSB-grown *S. auricularis* and *S. capitis* had almost equal proportions of SCFAs and BCFAs (*S. auricularis*, 47% SCFA, 53% BCFA; *S. capitis*, 50% SCFA, 49% BCFA) (Table 1), similar to *S. aureus* (51% SCFAs, 49% BCFAs). However, the other staphylococcal species grown in TSB had noticeably higher proportions of BCFAs (*S. epidermidis*, 66%; *S. hominis*, 67%; *S. haemolyticus*, 69%; *S. saprophyticus*, 76%). The BCFA fraction was composed of almost equal proportions of anteiso and iso fatty acids. 

In contrast to growth in TSB, the MHB-grown cultures had higher proportions of BCFAs (*S. auricularis*, 63%; *S. epidermidis*, 78%; *S. haemolyticus*, 79%; *S. hominis*, 82%; *S. saprophyticus*, 82%; *S. capitis*, 83%; and *S. aureus*, 81%) (Table 2). Anteiso fatty acids were markedly higher than the iso fatty acids, with a ratio of 1.49:1.00 anteiso:iso fatty acids in *S. auricularis* to as high as 3.10:1.00 in *S. epidermidis* in the BCFA fraction.

The different CONS showed a range of BCFAs to SCFAs ratios in TSB, with the Haemolyticus cluster group members (*S. haemolyticus* and *S. hominis*) having similar ratios to each other. *S. saprophyticus* had the highest proportion of BCFAs in either TSB or MHB-grown cultures. Growth in MHB increased the proportion of BCFAs in all the CONS. The differences in BCFAs to SCFAs ratios in MHB- versus TSB-grown cells have been attributed to the acquisition of branched-chain amino acids, which are the precursors of BCFAs [9], as free amino acids from MHB versus as peptides from TSB [7].

### 2.2. Growth in Serum Leads to Significant Amounts of Straight-Chain Unsaturated Fatty Acids in the Fatty Acid Profile

CONS grown in the common laboratory culture media lacked SCUFAs because they lack the genes for their biosynthesis [21]. However, the serum-grown organisms had significant proportions of SCUFAs (Table 3). Of the total fatty acids, the proportion of SCUFAs was about 25% in *S. auricularis*, 26% in *S. capitis*, 35% in *S. haemolyticus*, 55% in *S. hominis* and *S. saprophyticus*, and 64% in *S. epidermidis* compared to 41% in *S. aureus*. The proportion of BCFAs were significantly decreased in serum-grown CONS (Serum: 8%, TSB: 66% in *S. epidermidis*; Serum: 13%, TSB: 67% in *S. hominis*; Serum: 14%, TSB: 76% in *S. saprophyticus*; Serum: 26%, TSB: 69% in *S. haemolyticus*; Serum: 31%, TSB: 53% in *S. auricularis*; Serum: 35%, TSB: 50% in *S. capitis*; and Serum: 15%, TSB: 49% in *S. aureus*). The ratio of anteiso to iso fatty acids varied from about 1.02:1.00 in *S. epidermidis* to as high as 2.59:1.00 in *S. auricularis*. Importantly, BCFAs are not present in serum and, hence, the BCFAs are produced by biosynthesis [9]. 

Among the SCUFAs in the serum-grown CONS, linoleic acid (C18:2Δ9,12) was the most abundant component, comprising from 13% in *S. auricularis* to as high as 35% in *S. epidermidis* (Table 4). Oleic acid (C18:1Δ9) was the next most abundant SCUFA comprising as low as 8% in *S. auricularis* to as high as 19% in *S. epidermidis*. In contrast to these patterns, *S. capitis* had higher oleic acid content (11%) than linoleic acid (9%). Other SCUFAs detected in low amounts were arachidonic acid (C20:4Δ5,8,11,14), palmitoleic acid (C16:1Δ9), vaccenic acid (C18:1Δ11) and eicosenoic acid (C20:1Δ11). The serum-grown *S. aureus* had oleic acid (C18:1Δ9, 29%) as a major SCUFA fraction and 8% eicosenoic acid, and 3% C18:1Δ11 (Table 4). Interestingly, linoleic acid was not detected in serum-grown *S. aureus*.

The SCUFAs present in the total fatty acid profile of the CONS probably represent a mixture of SCUFAs truly incorporated into glycerolipids [7,19,22], and the association of serum lipids with the cell surface of the CONS. Lipidomic and ultrastructural analysis of *S. aureus* grown in the presence of human serum showed that serum lipids become associated with the cell surface and resist washing away with saline solution, but are removeable to a significant extent with Triton X-100 [22]. Kenanian et al. [21] showed that growth in serum allows *S. aureus* to bypass FASII inhibitors through the incorporation of fatty acids derived from host lipids. This is likely to be the case with the CONS also. These authors [21] reported significant incorporation of SCUFAs by *S. epidermidis* and *S. haemolyticus* in accordance with our findings.

### 2.3. Increased Membrane Fluidity in Mueller Hinton Broth- and Serum-Grown Coagulase-Negative Staphylococcal Species

To understand how the membrane biophysical properties of the different CONS were affected when grown in the different media, membrane fluidity was measured in terms of anisotropy values for each species. The higher the anisotropy values, the more rigid (less fluid) the membrane is. As shown in Table 5, the MHB and serum-grown CONS had significantly lower anisotropy values compared to those grown in TSB, indicating a more fluid membrane when grown in MHB consistent with higher portions of BCFAs in these cells. Noticeably, serum-grown cells had the lowest anisotropic values in all the species, indicating a large impact of SCUFAs on membrane fluidity. Among the TSB- or MHB-grown CONS, *S. saprophyticus* had most fluid membrane (0.156 ± 0.013 and 0.134 ± 0.006 respectively). On the other hand, the Haemolyticus cluster group members (*S. haemolyticus* and *S. hominis*) grown in TSB and MHB had the highest anisotropy values among the CONS. TSB-grown *S. hominis* had slightly higher (0.245 ± 0.029) anisotropy values than *S. haemolyticus* (0.228 ± 0.019), while serum-grown *S. hominis* had slightly lower (0.146 ± 0.027) anisotropy values than *S. haemolyticus* (0.169 ± 0.038). The Epidermidis group members (*S. epidermidis* and *S. capitis*) grown in TSB and MHB had comparable anisotropy values; however, the serum-grown *S. epidermidis* had a lower anisotropy value (0.096 ± 0.033). Notably, the serum-grown *S. epidermidis* had the highest proportion of SCUFAs and lowest anisotropy values among the CONS.

The CONS had lower anisotropy values than *S. aureus,* which indicates the membrane of the CONS were more fluid than those of *S. aureus*. This difference may be due to the presence of staphyloxanthin in the *S. aureus* membrane that is believed to decrease its fluidity [7,13,14]. This pigment is largely absent from CONS, although *S. hominis* cells had a yellowish pigment (this study and [35]), and this strain had the highest anisotropy values, indicating a less fluid membrane. Growth of the CONS in MHB increased BCFAs content and decreased anisotropy values. This is the same as what happens when carotenoid-deficient *S. aureus* or *Staphylococcus argenteus* are grown in MHB [15]. Increased membrane fluidity is expected to impact properties of the membrane such as permeability to various molecules including antimicrobial agents.

Interpretation of the anisotropy determinations in serum-grown cells is more difficult given the likely association of serum-lipids with the cell surface [22]. In cells grown in TSB or MHB, which do not have surface-associated lipids, we can be confident that 1,6-diphenyl-1,3,5-hexatriene (DPH) inserts into the cytoplasmic membrane and reports the status of the membrane. It is not clear where the DPH inserts in serum-grown cells, but may well insert in surface associated lipids and the cytoplasmic membrane [22].

### 2.4. Susceptibility of the Coagulase-Negative Staphylococcal Species Towards Host-Derived Antibacterial Fatty Acids Does Not Correlate with Their Fatty Acid Compositions

To understand the susceptibilities of the CONS towards known host-derived antibacterial fatty acids, the minimum inhibitory concentration (MIC) values of palmitoleic acid (C16:1Δ9), sapienic acid (C16:1Δ6), and linoleic acid (C18:2Δ9,12), which are all abundant on human skin, were determined against the different species of CONS. The CONS were more susceptible to sapienic acid (MICs, 32–64 µM) than to palmitoleic acid (MICs, 256–512 µM) (Table 6). *S. aureus* also had similar susceptibilities to sapienic acid (MICs, 32 µM) and palmitoleic acid (MICs, 256 µM) as the CONS.

*S. auricularis*, *S. capitis*, *S. saprophyticus*, and *S. aureus* were highly susceptible to linoleic acid (MICs, 32–64 µM), whereas *S. epidermidis*, *S. haemolyticus* and *S. hominis* were highly resistant to this antimicrobial fatty acid (MICs, >1024 µM). The antibacterial activities of these fatty acids involve insertion into and disruption of membrane function [24,25,26]. There were no obvious differences in fatty acid compositions that appeared to correlate with susceptibilities to the antimicrobial fatty acids. The lack of susceptibility of *S. epidermidis*, *S. haemolyticus* and *S. hominis* to linoleic acid may indicate the presence of an efflux pump, such as the inducible linoleic acid and arachidonic pump in some strains of *S. aureus* [36].

## 3. Materials and Methods 

*Staphylococcus auricularis* ATCC 33757^T^, *Staphylococcus capitis* ATCC 27840^T^, *Staphylococcus epidermidis* ATCC 12228^T^, *Staphylococcus haemolyticus* ATCC 27836^T^, *Staphylococcus hominis* ATCC 27844^T^, *Staphylococcus saprophyticus* ATCC 15305^T^, and *Staphylococcus aureus* RN450 were grown in various culture media and analyzed for growth characteristics through the determination of culture turbidity [7], fatty acid composition and membrane fluidity. The growth media used were BactoTM tryptic soy broth (TSB; Becton, Dickinson and Company, Sparks Glencoe, MD, USA), BactoTM Mueller–Hinton broth (MHB; Becton, Dickinson and Company, Sparks Glencoe, MD, USA) and 100% human serum (BioreclammationIVT, Westbury, NY, USA). MHB was supplemented with 25 mg/liter Ca^2+^ and 12.5 mg/liter Mg^2+^. Human serum was heated to 56 °C for 30 min to inactivate complement before using. Staphylococci were grown in 50 mL of medium in a 250 mL Erlenmeyer flask with shaking at 200 rpm at 37 °C unless otherwise specified.

The different species were grown to an optical density (OD_600_) of about 0.8 in TSB, MHB and 100% human serum, harvested by centrifugation at 2000× *g* for 5 min, followed by washing two times with cold 1x phosphate buffer saline (PBS; 8 g/L NaCl, 0.2 g/L KCl, 1.44 g/L Na_2_HPO_4_ and 0.24 g/L KH_2_PO_4_, pH 7.4). The samples were sent for fatty acid methyl ester analysis by gas-liquid chromatography at Microbial ID, Inc. (Newark, DE, USA) and identified by the MIDI microbial identification system (Sherlock 4.5 microbial identification system, Microbial ID Inc., Newark, DE, USA), as described previously [7]. The value of the percent composition of an individual fatty acid typically has a reproducibility of ±0.02 to ±1.44 standard error of the mean (SEM) in extensive experience in our laboratory [7,15]. 

The different strains were grown as described for the fatty acid analysis and membrane fluidities of the cells were determined using DPH (Sigma-Aldrich, Cleveland, OH, USA) as described previously [37]. Briefly, DPH stock solution (10 mM) was prepared in tetrahydrofuran (Sigma-Aldrich, Cleveland, OH, USA), vortexed for 10 min and working solution (10 µM) was prepared in PBS followed by vortexing for 10 min. The cell pellets were washed twice with cold 1x PBS (pH 7.4) and resuspended to an OD600 of ~0.75, DPH dye was added at 5 µM concentration, the tube was wrapped with aluminum foil and incubated at 30 °C in a water bath. All steps involving DPH were carried out in the dark. Fluorescence polarization emitted by the fluorophore was measured using a PTIModel Quanta Master-4 Scanning Spectrofluorometer at an excitation wavelength of 360 nm and an emission wavelength of 430 nm. The experiments were performed with three separate fresh batches of the cultures. 

MICs of known antimicrobial fatty acids, viz., palmitoleic acid (C16:1Δ9, Larodan Fine Chemicals, Solna, Sweden), sapienic acid (C16:1Δ6, Larodan Fine Chemicals, Solna, Sweden), and linoleic acid (C18:2Δ9,12; Larodan Fine Chemicals, Solna, Sweden), dissolved in absolute ethanol, were determined against the staphylococcal species by the broth microdilution method as recommended by the Clinical Laboratory Standards Institute [10] guidelines. Briefly, serial dilutions of each antimicrobial fatty acid were prepared in a microtiter plate with 90 µL of TSB containing defined concentrations of the fatty acids for 100 µL total volume. Each strain was grown to OD600 of 1.0 at 37 °C with shaking at 200 rpm, diluted 20 times in sterile TSB, and 10 µL of the cell suspension was inoculated in each well in triplicate on the microtiter plate. The plates were incubated at 37 °C for 16 h and the MIC was determined as the concentration of the fatty acid that inhibited growth of the given species. At least three independent microtiter plate assays were carried out for each species. 

## 4. Conclusions

The fatty acid compositions of several CONS showed a similar growth-environment dependent plasticity as previously established for *S. aureus*. The growth medium used for routine antibiotic susceptibility testing, cation-supplemented MHB dramatically increased the proportions of BCFAs and membrane fluidity compared to cells grown in TSB. Preformed SCUFAs become prominent members of the total fatty acid profile in cells grown in serum, indicating a lowered dependency on fatty acid biosynthesis. The different species were susceptible to sapienic acid but showed varying susceptibilities to palmitoleic acid and linoleic acid.

Our findings suggest that agents directed against the FASII system are unlikely to be of value against CONS given the ability of these organisms to incorporate host fatty acids into their lipids. In the context of these organisms as inhabitants of the skin, their ability to incorporate major amounts of linoleic acid into their lipids may suggest that this is a means of detoxifying this antimicrobial fatty acid. Further study of the incorporation of individual fatty acids is probably warranted for the CONS in the context of the skin environment and their exposure to host lipids. 

## Figures and Tables

**Table 1 antibiotics-09-00214-t001:** Fatty acid composition (%) of Tryptic Soy Broth-grown coagulase-negative staphylococcal species.

Fatty Acids		Cluster Groups					
		Auricularis	Epidermidis		Haemolyticus		Saprophyticus
	*S. aureus*	*S. auricularis*	*S. capitis*	*S. epidermidis*	*S. haemolyticus*	*S. hominis*	*S. saprophyticus*
SCFA ^1^	50.7	47.1	50.0	34.1	30.6	32.2	23.4
IFA_odd_ ^2^	17.0	20.2	19.3	25.6	26.8	27.9	38.4
IFA_even_ ^3^	4.1	6.6	4.4	3.1	5.1	3.9	2.7
AFA ^4^	28.2	25.9	25.6	36.9	37.0	35.5	35.2
SCUFA ^5^	0	0	0	0	0	0	0
Total BCFAs	49.3	52.7	49.3	65.6	68.9	67.3	76.3
BCFA:SCFA	0.97	1.12	0.99	1.93	2.25	2.09	3.25
AFA:IFA	1.34	0.97	1.08	1.29	1.16	1.12	0.86

^1^ SCFA, straight-chain fatty acid; ^2^ IFA_odd_, iso fatty acid with odd number of carbon atoms; ^3^ IFA_even_, iso fatty acid with even number of carbon atoms; ^4^ AFA, anteiso fatty acid; ^5^ SCUFA, straight-chain unsaturated fatty acid; BCFAs, branched-chain fatty acids; BCFAs comprise IFA_odd_, IFA_even_, and AFA.

**Table 2 antibiotics-09-00214-t002:** Fatty acid composition (%) of Mueller–Hinton broth-grown coagulase-negative staphylococcal species.

Fatty Acids		Cluster Groups					
		Auricularis	Epidermidis		Haemolyticus		Saprophyticus
	*S. aureus*	*S. auricularis*	*S. capitis*	*S. epidermidis*	*S. haemolyticus*	*S. hominis*	*S. saprophyticus*
SCFA ^1^	19.0	37.2	16.5	22.0	20.6	18.0	17.9
IFA_odd_ ^2^	15.3	14.7	21.6	16.7	21.1	19.5	24.5
IFA_even_ ^3^	4.0	10.5	2.2	2.3	2.3	2.9	2.5
AFA ^4^	61.7	37.6	59.3	59.0	56.0	59.4	55.2
SCUFA ^5^	0	0	0	0	0	0	0
Total BCFAs	81.0	62.8	83.1	78.0	79.4	81.8	82.2
BCFA:SCFA	4.26	1.69	5.05	3.54	3.86	4.56	4.59
AFA:IFA	3.12	1.49	2.49	3.10	2.39	2.65	2.05

^1^ SCFA, straight-chain fatty acid; ^2^ IFA_odd_, iso fatty acid with odd number of carbon atoms; ^3^ IFA_even_, iso fatty acid with even number of carbon atoms; ^4^ AFA, anteiso fatty acid; ^5^ SCUFA, straight-chain unsaturated fatty acid; BCFAs, branched-chain fatty acids; BCFAs comprise IFA_odd_, IFA_even_, and AFA.

**Table 3 antibiotics-09-00214-t003:** Fatty acid composition (%) of human serum-grown coagulase-negative staphylococcal species.

Fatty Acids		Cluster Groups					
		Auricularis	Epidermidis		Haemolyticus		Saprophyticus
	*S. aureus*	*S. auricularis*	*S. capitis*	*S. epidermidis*	*S. haemolyticus*	*S. hominis*	*S. saprophyticus*
SCFA ^1^	43.9	44.4	39.9	28.1	39.1	32.0	30.7
IFA_odd_ ^2^	3.2	4.7	9.3	2.4	6.5	3.9	4.0
IFA_even_ ^3^	1.0	3.9	0.5	1.5	1.1	1.3	0.8
AFA ^4^	11.1	22.4	24.7	4.0	18.4	8.0	9.6
SCUFA ^5^	40.9	24.6	25.6	63.9	34.9	54.8	55.0
Total BCFAs	15.3	31.0	34.5	7.9	26.0	13.2	14.4
BCFA:SCFA	0.35	0.70	0.86	0.28	0.66	0.41	0.47
AFA:IFA	2.64	2.59	2.52	1.02	2.44	1.55	1.99
SCUFA:SCFA	0.93	0.55	0.64	2.27	0.89	1.71	1.79

^1^ SCFA, straight-chain fatty acid; ^2^ IFA_odd_, iso fatty acid with odd number of carbon atoms; ^3^ IFA_even_, iso fatty acid with even number of carbon atoms; ^4^ AFA, anteiso fatty acid; ^5^ SCUFA, straight-chain unsaturated fatty acid; BCFAs, branched-chain fatty acids; BCFAs comprise IFA_odd_, IFA_even_, and AFA.

**Table 4 antibiotics-09-00214-t004:** Proportion (%) of the mono-, di-, and poly-unsaturated fatty acids in the coagulase-negative staphylococcal species grown in human serum.

SCUFA ^1^		Cluster Groups					
		Auricularis	Epidermidis		Haemolyticus		Saprophyticus
	*S. aureus*	*S. auricularis*	*S. capitis*	*S. epidermidis*	*S. haemolyticus*	*S. hominis*	*S. saprophyticus*
C16:1Δ9	0.5	1.3	1.0	2.7	1.1	2.2	2.3
C18:1Δ11	2.9	0.9	1.6	2.2	1.7	2.0	1.9
C18:1Δ9	29.0	8.3	11.1	18.7	11.8	16.6	16.9
C20:1Δ11	8.2	0	2.3	1.5	4.7	2.7	1.0
C18:2Δ9,12	0	12.6	8.6	35.2	14.0	28.2	29.4
C20:4Δ5,8,11,14	0.4	1.5	1.0	3.3	1.4	2.8	3
Total SCUFA	41.0	24.6	26.6	63.6	34.7	54.5	54.5
Di:Mono ratio	0	1.20	0.54	1.40	0.72	1.20	1.33

^1^ SCUFA, straight-chain unsaturated fatty acids

**Table 5 antibiotics-09-00214-t005:** Membrane fluidity (anisotropy values ^1^) of the coagulase-negative staphylococcal species grown in tryptic soy broth (TSB), Mueller–Hinton broth (MHB) and serum.

Cluster Group	Staphylococci	TSB	MHB	Serum
	*S. aureus*	0.315 ± 0.025	0.318 ± 0.035	0.212 ± 0.017 ***
Auricularis	*S. auricularis*	0.196 ± 0.008	0.166 ± 0.018 **	0.114 ± 0.003 ***
Epidermidis	*S. capitis*	0.186 ± 0.015	0.152 ± 0.015 **	0.110 ± 0.002 ***
	*S. epidermidis*	0.180 ± 0.008	0.145 ± 0.013 **	0.096 ± 0.033 ***
Haemolyticus	*S. haemolyticus*	0.228 ± 0.019	0.188 ± 0.029 **	0.169 ± 0.038 ***
	*S. hominis*	0.245 ± 0.029	0.183 ± 0.028 **	0.146 ± 0.027 ***
Saprophyticus	*S. saprophyticus*	0.156 ± 0.013	0.134 ± 0.006 *	0.118 ± 0.008 ***

^1^ Statistical significance between the anisotropy values of the TSB- and MHB-grown CONS, and the TSB- and serum-grown CONS species was calculated by two-tailed Student *t*-test (α = 0.05). * *p* ≤ 0.05; ** *p* ≤ 0.01; *** *p* ≤ 0.001.

**Table 6 antibiotics-09-00214-t006:** Minimum inhibitory concentrations (µM) of some antimicrobial fatty acids.

Cluster Group	Staphylococci	Sapienic Acid (C16:1Δ6)	Palmitoleic Acid (C16:1Δ9)	Linoleic Acid (C18:2Δ9,12)
	*S. aureus*	32	256	64
Auricularis	*S. auricularis*	64	256	32
Epidermidis	*S. capitis*	32	256	32
	*S. epidermidis*	64	512	2048
Haemolyticus	*S. haemolyticus*	64	512	2048
	*S. hominis*	64	256	1024
Saprophyticus	*S. saprophyticus*	32	256	32

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
