# Peer review of "Plasticity of Coagulase-Negative Staphylococcal Membrane Fatty Acid Composition and Implications for Responses to Antimicrobial Agents"

_antibiotics, 2020, doi:10.3390/antibiotics9050214_

Round 1

Reviewer 1 Report

Manuscript ID: antibiotics-772243

Plasticity of Coagulase-Negative Staphylococcal Membrane Fatty Acid Composition and Implications for Responses to Antimicrobial Agents

In this work, Tiwari, Gatto and Wilkinson studied the fatty acid (FA) composition and the cell membrane response of different species of coagulase-negative staphylococci (CONS) to different environments. These species are common skin commensals, but that are also human pathogens and some of them are resistant to several antibiotics. To this end, the fatty acid profile was evaluated as an indicator of the bacterial membrane plasticity to different growth media. Six strains of CONS were used. The coagulase-positive S. aureus that was used as a reference. The bacteria were grown in different culture media: TSB, MHB supplemented with cations, and human serum. The FA profile of the bacteria was determined, the membrane fluidity characteristics were analyzed through fluorescence polarization, and the minimum inhibitory concentrations of antibacterial FA (C16:1n-9, C16:1n-6 and C18:2n-6) were determined in the different strains using the microdilution plate method. It has been shown a plasticity of membrane FA dependent on the growth environment in which the cation-supplemented MHB medium increases the proportion of branched-chain FA (BCFA) and the membrane fluidity compared to TSB, which also depends on the prior biosynthesis of the FA. On the contrary, bacterial grown in serum makes straight-chain unsaturated FA more abundant, which shows less recourse to the biosynthesis of new FA. Sapienic acid and in some cases linoleic acid have shown inhibitory activity on CONS strains. One of the major findings from this study is that preformed FA can replace biosynthesized FA in the glycerolipids of CONS, and that bacterial FA biosynthesis system II may not be a good target for antimicrobial agents in these organisms.

This work is interesting for a large audience from the biological, biochemical and biomedical sciences. Overall, the manuscript is well written, the text is supported by correct and pertinent bibliography, the “Introduction” is well structured, the results are clear and concisely explained, as well as the methods. However, I have some remarks.

The Introduction should be complemented with some information about branched-chain fatty acids (FA) since nothing is said about the iso- and ante-iso FA before they appear in the Results’ section. The experimental design is well defined, but some more detail could be provided in the Material and Methods’ section, namely the inclusion of  “Statistical Analysis”. This is not only to support the membrane fluidity results but also it should be performed for the other evaluated parameters. Also, some results can be more discussed in light of the scientific literature, as the importance of some of the findings, and if they agree with the previous reports or not. Most importantly, I think that the authors should reformulate their Conclusions’ section. In fact, the major conclusions pointed out by the authors, are they main findings. But, a final paragraph should be added stating how these discoveries can be important to better known the microbial skin environment/protection, virulence of CONS, and/or antimicrobial susceptibility/targets to these species.

This work has scientific merit and fits in the scope of “Antibiotics”. However, it should be reviewed before being considered for publication. Please see specific comments below.

Comments:

  1. Lines 30-31: Keywords: Try to avoid repeating words from the title. You may use up to ten pertinent keywords specific to the article, other than those used in the title since on-line searching engines will go through title and keywords to find the articles. Other examples of keywords to be considered: branched-chain fatty acid; straight-chain fatty acid; fluidity; antimicrobial susceptibility; membrane lipid.
  2. Italics for genus or species names: lines 34, 202, 344, 353, 358, 359.
  3. Line 80: “Very closed related structures” refers to fatty acids? Please clarify.
  4. Lines 85-87: Is this the question or one of the questions that this study aims to answer? This should be highlighted in the end of the Introduction.
  5. Line 95: Reference #31: Can you provide a more recent reference?
  6. Lines 103-104: “differences in the proportions of individual fatty acids amongst individual species (32)”. In my opinion, this sentence is not clear to the reader. Please, clarify. Also, can you provide a more recent reference for this idea?
  7. Lines 104-108: This part should be in a new paragraph. It is also confusing because there is a mixture of author’s findings but there is no aim of the study clearly defined at the end of the Introduction. This paragraph should be replaced or inserted after the aim of the study.
  8. Line 117 and others in the “Results and Discussion” section: the percentages of BCFA are presented throughout the text as the sum of iso- and ante-iso fatty acids from the different tables. I think that this is not intuitive for the readers and a new line should be inserted in the tables with the sum of the BCFA, that will correspond to the values referred in the text. Also, in the tables, it should be indicated that IFAodd, IFAeven and AFA are BCFA. Either in the table legend or in the table itself. Or both. Please do this for every table.
  9. Line 121: “the MHB-grown cultures has significantly higher proportions” – the term “significant” can only be used when statistical tests are performed. Why did the authors not performed statistical analysis for the data from Tables 1 to 4 and 6? I think this is a major weakness of this study.
  10. Line 124: “with a ratio of 1.49” – What does this ratio mean? Why is it important? It should be clarified in the text.
  11. Line 121-125: This part of the text should be after Table 1.
  12. Line 126 and others: Abbreviations in table titles: TSB, MHB, CONS. I suggest writing these terms in full instead of the abbreviations.
  13. Lines 127-128 and the other table legends: The presented values are means from how many experiments? No standard error of the mean is shown, but is should be there. Please consider performing statistical analysis to be sure of significant differences among species.
  14. Branched-chain fatty acids: Odd and even iso and ante-iso fatty acids are not mentioned in the Introduction. They are suddenly mentioned in the Results and Discussion section. I think that something should be briefly explained in the Introduction concerning these fatty acids: characteristics, differences, importance within the context/aim of this work.
  15. Line 140: Please avoid abbreviations in the headings and subheadings.
  16. Lines 141-142: Please provide a reference for this idea (first sentence of the paragraph).
  17. Line 149: Again, why is this ratio important?
  18. Line 150: “BCFAs are produced by biosynthesis.” Please provide a reference for this.
  19. Line 151: Title of table 3: “Fatty acid composition ‘(%)’ of human serum…”
  20. Lines 156-157: Is this in agreement in the literature? Please comment and cite the appropriate references.
  21. Lines 157-158: Can the authors explain why capitis has a higher oleic than linoleic acid content? And for the rest of the paragraph, how do these findings agree with the literature? Please provide your comments supported by references.
  22. Line 163 (Table 4): It would be good to have the sum of the SCUFA in the table for easiness of interpretation, as it is in Table 3.
  23. Line 173: Is this in agreement with the authors’ results?
  24. Lines 194-195: Statistical analysis should be described in a new paragraph in the Materials and Methods section.
  25. Lines 197-200: Understanding the relationship between anisotropy and membrane fluidity/rigidity may not be straightforward to the reader in this part of the text. Please clarify. “The CONS has lower anisotropy values that aureus”. This means that the membranes of CONS are more fluid or less rigid than that of S. aureus? Then, “This ‘difference’ may be due to…”.
  26. Line 207: DPH appears firstly in the text in this line. Please provide the meaning of this abbreviation (that is already in the Methods part).
  27. Line 209: Please provide a reference for this sentence.
  28. Section 3. Materials and Methods section – I think that it would be better to the reader if you could make paragraphs and detach the subheadings.
  29. Line 233-234: “analyzed for growth characteristics” – How was this evaluated?
  30. Line 238: Please explain why the serum was heated before use.
  31. Line 239: How many biological replicates were tested for each strain/condition? Please mention that in the text and in the table legends.
  32. Line 245: Please briefly explain the principle of the method. Does it use GC analysis? Of any hyphenated GC technique for fatty acid profiling?
  33. Line 260: “dissolved in ethanol”: In pure ethanol? What was the proportion of ethanol in the mixture?
  34. Conclusions: Please improve your conclusion by answering to these questions: How can these results translate into a real skin environment? What is the major conclusion (novelty) of this study? What is the impact of the findings in the current knowledge on CONS?

Reviewer 2 Report

The manuscript “Plasticity of Coagulase-Negative Staphylococcal Membrane Fatty Acid Composition and Implications for Responses to Antimicrobial Agents” by K. Tiwari and co-authors is mainly dedicated to the determination of membrane fatty acid composition of CoNS grown in different media. The authors demonstrate that the amount of branched fatty acids increases when different CoNS species are cultivated in MHB compared to TSB, and so does membrane fluidity. Furthermore, the authors provide the data in support of the hypothesis that CoNS can incorporate host-derived unsaturated fatty acids into their membranes, also increasing membrane fluidity. Finally, the authors provide the data about susceptibility of different CoNS species to several antimicrobial fatty acids.

The manuscript is well written, the results are presented in a clear and concise fashion and fully support the conclusions. I read the manuscript with great interest and I think it can be accepted in the present form. However, I do have a couple of remarks that might improve the manuscript:

Line 39: “of animal host-associated to environmental species” => “from animal host-associated to environmental species”?

Line 237: cation-adjusted MHB usually has ~25 mg/L calcium and ~12.5 mg/L magnesium. Here, the same numbers are given for the concentrations in mM, not mg/L. Is this a typo or was such a high amount of calcium and magnesium used?

Table 5. Ideally, correction for multiple comparisons should be applied to p-values. In this table the authors make 14 comparisons (7 species, two comparisons each) and assess p-values for all of them. This raises the possibility of experiment-wise error, which can be alleviated by a Bonferroni or Dunn-Sidak correction. It is unlikely to change much of the conclusions, but could be done for the sake of good practice.

Line 202: Staphylococcus argenteus should be italicized.

Tables 1-4 provide the data on fatty acid compositions. As I understand, these experiments were only done once, since no standard errors or deviations are provided. Thus, the variability in the fatty acids composition between bacterial culture batches is unclear. How do these differences compare to the typical variability between different batches? Are the differences between species and media significant?

Round 2

Reviewer 1 Report

Manuscript ID: antibiotics-772243

The authors have submitted a revised version of the above-mentioned manuscript following the reviewers’ remarks. I think it has improved and can now be accepted for publication in the present form.